# Immune Cells in the Spleen of Mice Mediate the Inflammatory Response Induced by *Mannheimia haemolytica* A2 Serotype

**DOI:** 10.3390/ani14020317

**Published:** 2024-01-19

**Authors:** Zizhuo Jiao, Junming Jiang, Yong Meng, Guansheng Wu, Jiayang Tang, Taoyu Chen, Yujing Fu, Yuanyuan Chen, Zhenxing Zhang, Hongyan Gao, Churiga Man, Qiaoling Chen, Li Du, Fengyang Wang, Si Chen

**Affiliations:** Hainan Key Lab of Tropical Animal Reproduction, Breeding and Epidemic Disease Research, Animal Genetic Engineering Key Lab of Haikou, School of Tropical Agriculture and Forestry, Hainan University, Haikou 570228, China; jiaozizhuo1998@outlook.com (Z.J.); jmm992847@163.com (J.J.); alpha.meng@outlook.com (Y.M.); wgs2192208700@163.com (G.W.); 18309839751@163.com (J.T.); 18889542406@163.com (T.C.); m15383468682_1@163.com (Y.F.); chen727501398@163.com (Y.C.); zxzhang23@163.com (Z.Z.); gaohongyan@hainanu.edu.cn (H.G.); manchuriga@163.com (C.M.); chenqiaoling@hainanu.edu.cn (Q.C.); kych2008dl@163.com (L.D.)

**Keywords:** *Mannheimia haemolytica*, spleen, NF-κB, cell adhesion molecule, hematopoietic cell lineage

## Abstract

**Simple Summary:**

*Mannheimia haemolytica* (*M. haemolytica*) is a significant bacterial pathogen that causes substantial economic losses to the livestock industry by infecting ruminants. In our previous study, we successfully isolated and identified a strain of *M. haemolytica* from Hainan Black goats. However, the pathogenicity of this strain and the molecular mechanism underlying the interaction between *M. haemolytica* and animals have not been investigated. In this study, we constructed a *M. haemolytica* infection model in mice to analyze the histopathology of the bacterial infection and explore the immune mechanisms employed by the murine spleen to resist *M. haemolytica* infection.

**Abstract:**

(1) Background: *Mannheimia haemolytica* (*M. haemolytica*) is an opportunistic pathogen and is mainly associated with respiratory diseases in cattle, sheep, and goats. (2) Methods: In this study, a mouse infection model was established using a *M. haemolytica* strain isolated from goats. Histopathological observations were conducted on various organs of the mice, and bacterial load determination and RNA-seq analysis were specifically performed on the spleens of the mice. (3) Results: The findings of this study suggest that chemokines, potentially present in the spleen of mice following a *M. haemolytica* challenge, may induce the migration of leukocytes to the spleen and suppress the release of pro-inflammatory factors through a negative feedback regulation mechanism. Additionally, an interesting observation was made regarding the potential of hematopoietic stem/progenitor cells congregating in the spleen to differentiate into immune cells, which could potentially collaborate with leukocytes in their efforts to counteract *M. haemolytica* invasion. (4) Conclusions: This study revealed the immune regulation mechanism induced by *M. haemolytica* in the mouse spleen, providing valuable insights into host–pathogen interactions and offering a theoretical basis for the prevention, control, and treatment of mannheimiosis.

## 1. Introduction

*Mannheimia haemolytica* (*M. haemolytica*), previously known as *Pasteurella haemolytica*, is a facultative anaerobic, gram-negative bacterium that belongs to the genus Mannheimia [1]. The bacterium is short, rod-shaped, possesses a capsule and pili, and is non-spore-forming. Wright staining shows bipolar staining, and Gram staining results in a red color [2]. *M. haemolytica* is an opportunistic pathogen that colonizes and persists in the nasopharynx, respiratory mucosal surface, tonsils, and other tissues of ruminants. It is capable of evading the host’s immune response [3]. *M. haemolytica* is a part of the natural bacterial community in the upper respiratory tract of ruminants [4,5]. Only transportation, environmental stress, or viral infection can weaken the animal’s immune system, rendering it more susceptible to severe respiratory diseases caused by this bacterium and allowing *M. haemolytica* to colonize the lungs [6,7].

*M. haemolytica* is currently classified into 12 serotypes based on the results of experiments conducted by Angen et al. [8]. Serotypes A1, A2, and A6 are the most commonly isolated serotypes worldwide [9], with serotype A2 being the primary pathogen in sheep and goats [10]. *M. haemolytica* is an acute infectious pathogen that can cause sepsis and tissue hemorrhaging [11]. It is one of the most important respiratory pathogens in domestic ruminants. It causes severe outbreaks of acute pneumonia in cattle and goats, such as fibro-necrotizing pneumonia [12], as well as septicemia in young animals and mastitis in sheep [13]. Various livestock diseases caused by *M. haemolytica* often cause significant economic losses due to morbidity, mortality, feed efficiency, reduced animal production, and increased treatment costs [14].

The lung lesions caused by *M. haemolytica* typically manifest as anteroventral solid lesions, exhibiting a darkened color and an overall dark reddish or greyish-brown discoloration of the lungs. Additionally, there is observed thickening of the interlobular septa, fibrin adhesion on the pleura, and the presence of fibrinous pleurisy; the lungs exhibit extensive thrombus formation and the presence of necrotic foci, which are discretely distributed. Localized bronchitis and bronchiolitis were observed [15,16]. In chronic pneumonia, the necrotic area was infiltrated by macrophages and fibroblasts and wrapped in the capsule of fibrous trabeculae. Over time, these areas were remodeled as granulomatous areas [12]. The pathological manifestations of mastitis caused by *M. haemolytica* include ecchymosis and ecchymotic bleeding, accompanied by neutrophil infiltration and, in advanced stages, congestion, hemorrhage, venous thrombosis, and mammary parenchymal necrosis; the mammary ducts are full of fibrin, milk clots, and inflammatory exudates [17]. The reproductive system function can be influenced in animals following infection with *M. haemolytica*, as indicated by several studies [18,19].

The virulence factors of *M. haemolytica* mainly include capsule [20], lipopolysaccharide (LPS) [21], leukotoxin (LKT) [22], adhesins [23], outer membrane proteins [24], and proteases [25]. Among them, LPS and LKT are closely related to the pathogenesis of *M. haemolytica*. Currently, the molecular pathogenesis of *M. haemolytica* is studied as follows. LPS is an important virulence factor of *M. haemolytica*. As the main component of the cell wall of Gram-negative bacteria, the lipid A of LPS has endotoxin activity. LPS can facilitate *M. haemolytica* to evade the immune barrier of the host to proliferate in the lungs, induce an inflammatory cytokine response, lysing neutrophils and alveolar macrophages, and ultimately enhance lung damage. The LPS-CD14 complex can initiate intracellular signals, such as the NF-κB signaling pathway, by interacting with another membrane protein, Toll-like receptor 4 (TLR4), which in turn induces the release of pro-inflammatory cytokines [15]. LKT, a member of the bacterial repeats-in-toxin (RTX) family, is the most important virulence factor of *M. haemolytica*. LKT is active through a specific interaction with the β2 integrin LFA-1 [26]. Low concentrations of LKT may allow bacteria to evade host immune surveillance by destroying macrophages and neutrophils in the intrinsic response and enhancing the inflammatory process. On the other hand, high concentrations of LKT cause cell necrosis, which leads to lesions. In addition, KLT and LPS can form complexes that act on various pathways to form a complex network that controls cell survival and apoptosis [27]. LPS can also activate mononuclear phagocytes in the absence of LPS-binding protein (LBP), suggesting that the CD14-dependent pathway may not be the only way for LPS to interact with macrophages and monocytes [15].

To investigate the pathological changes and immune defense mechanisms of the spleen, the largest secondary lymphoid organ in the body, RNA-sequencing (RNA-seq) analysis was used to identify changes in spleen signal transduction and immune pathways in mice after infection. The aim of this study was to elucidate the immune regulation mechanism of the spleen in resisting the invasion and infection of *M. haemolytica* and provide valuable insights into the function and immune mechanism of the spleen. The findings of this study may also contribute to the development of the *M. haemolytica* infection model.

## 2. Materials and Methods

### 2.1. Ethical Review

The experimental procedure of this study was approved by the Academic Committee of Hainan University, following Animal Welfare and Ethics, with the approval number HNUAUCC-2023-00206.

### 2.2. Bacteria Source and Serotypes Identification

*M. haemolytica* was isolated from a nasal swab of Hainan black goat and named Mh-HN0721, which was preserved in our laboratory. Single colonies were then selected from Tryptic Soya Agar (TSA) medium (Qingdao Haibo Biotechnology Co., Ltd., Qingdao, China) for serotyping of Mh-HN0721 (Table 1) [28,29]. The PCR reaction system was 50 μL, consisting of 25 μL Taq Master mix, 2 μL upstream primer, 2 μL downstream primer, and 21 μL ddH_2_O. Single colonies were picked and mixed with the system. The reaction was carried out in an S1000 thermal cycler (BIO-RAD, Hercules, CA, USA). The reaction procedure was started at 95 °C for 15 min, followed by 35 cycles at 94 °C for 30 s, 55 °C for 45 s, 72 °C for 1 min, and ended at 72 °C for 10 min. After amplification, the PCR products were electrophoresed on 1% (*w*/*v*) agarose gel.

### 2.3. Experimental Animal

Thirty healthy specific-pathogen-free (SPF) two-week-old male Kunming mice (18–22 g) were purchased from Guangzhou Yancheng Biotechnology Co., Ltd. (Guangzhou, China). The animal license number was SYXK (Qiong) 2023-0031. The mice were placed in ventilated sterile cages with free access to food and water. These cages were in a separate room with a constant temperature of 24 °C. Upon arrival, the mice had 1 day for adaptation and were then used in this study. The mice were randomly divided into a challenge group (*n* = 21) and a control group (*n* = 9). In the challenge group, mice were euthanized by cervical dislocation at 9, 18, and 27 h post infection (h.p.i), respectively. As for the control group, the mice were euthanized directly before Mh-HN0721 challenge (0 h.p.i) (Figure 1). In each time point, 3 mice were used for the determination of bacterial load and 3 mice were used for hematoxylin and eosin staining (H&E staining). Moreover, another 3 mice in the control group and 3 mice in the challenge group (18 h.p.i) were euthanized directly. Their spleens were collected for RNA-sequencing (RNA-seq) using the Illumina platform (NovaSeq 6000, Illumina, San Diego, CA, USA) at Majorbio (Majorbio, Shanghai, China).

### 2.4. Culture and Pathogenicity of Mh-HN0721

The Mh-HN0721 glycerol stock in −80 °C was thawed and streaked onto TSA medium. After culture at 37 °C for 24 h, the single colonies on TSA medium were picked and inoculated in 15 mL tryptic soy broth (TSB) medium (Qingdao Haibo Biotechnology Co., Ltd., Qingdao, China), which was then cultured in a thermostatic oscillation incubator (HZQ-X100A, Shanghai Yiheng Scientific Instrument Co., Ltd., Shanghai, China) at 37 °C and 180 g/min for 24 h. Optical density (OD_600_) value and corresponding bacterial quantity of the inoculated TSB medium of the bacterial solution were measured at the indicated time points. Meanwhile, Gram staining was conducted to observe the morphology of Mh-HN0721. The bacterial solution was centrifuged at 4500 rpm for 5 min. Subsequently, the supernatant was discarded, and the pellet was resuspended in phosphate buffer solution (PBS). Each mouse in the challenge group was intraperitoneally challenged with 0.2 mL bacterial solution, containing 5 × 10^9^ colony forming units (CFUs) of Mh-HN0721. For the control group, each mouse was intraperitoneally challenged with 0.2 mL PBS. The mental and behavioral statuses of mice were observed and documented.

### 2.5. Bacterial Load of Mh-HN0721 in Mice Spleen

To determine the bacterial load in the spleen of mice following Mh-HN0721 challenge, 9 mice from the challenge group were euthanized at 9, 18, and 27 h.p.i, respectively. Each designated time point contained 3 biological replicates. After removing excess fat and connective tissues, the spleens were promptly collected and placed in a 2 mL sterile Eppendorf (EP) tube (Wuhan Servicebio Biotechnology Co., Ltd., Wuhan, China) with three 3 mm grinding beads. An appropriate amount of PBS was supplemented in each tube to equalize their weights (Appendix A). The EP tubes were placed in a tissue grinder and ground for 2 min. Subsequently, the splenic homogenates were serially diluted to 10-fold dilutions using PBS. Diluents of different concentration were coated onto TSA medium thrice. The plates were observed and recorded after 24 h of culture in a constant temperature incubator (HZQ-X100A, Shanghai Yiheng Scientific Instrument Co., Ltd., Shanghai, China) set at 37 °C. Afterwards, the bacterial loads of Mh-HN0721 in mice spleens were calculated by the plate-counting method.

### 2.6. Histopathological Examination of the Spleen

To investigate the pathological changes in mice organs caused by *M. haemolytica* infection over time, 3 mice from the challenge group were euthanized at 18 h.p.i, respectively. The entire heart, liver, spleen, lung, and kidney were promptly collected and placed in a tissue-fixation solution (Wuhan Servicebio Biotechnology Co., Ltd., Wuhan, China) for over 48 h. The fixed tissues were dehydrated by graded ethanol, transparentized by xylene, dipped in wax, and then embedded in paraffin. The tissues were cut into 4 μm-thick sections using a microtome (Leica RM2245, Wetzlar, Germany), after which they were dewaxed and stained with H&E (Zhuhai Baso Biotechnology Co., Ltd., Zhuhai, China). The sections were dehydrated, sealed, and observed under a microscope. To investigate the pathological changes in mice spleens caused by *M. haemolytica* infection over time, 9 mice from the challenge group were euthanized at 9, 18, and 27 h.p.i, respectively. Each designated time point contained 3 biological replicates. Three mice in the control group were euthanized as controls. The above experimental procedures were repeated and observed under a microscope [30,31].

### 2.7. RNA Preparation, Library Construction, and Sequencing

According to body status and splenic histopathological changes, 3 mice in the worst condition (18 h.p.i) from the challenge group were euthanized to explore the interaction between *M. haemolytica* and the host immune system at the gene expression level. Meanwhile, 3 mice from the control group were used as controls. All 6 spleens were immediately collected and preserved in liquid nitrogen upon euthanasia for subsequent RNA-seq. Additionally, DNA of the 3 spleens from the challenge group were extracted, and specific PCR was conducted as described by Alexander TW et al. [32] to demonstrate the existence of *M. haemolytica* after challenge. The sequences of bacterial primer were F: 5′-GCA GGA GGT GAT TAT TAA AGT GG-3′, R: 5′-CAG CAG TTA TTG TCA TAC CTG AAC-3′. The length of the amplified product was 206 bp. The PCR reaction system was 50 μL, consisting of 25 μL Taq Master mix, 2 μL upstream primer, 2 μL downstream primer, and 21 μL ddH_2_O. Single colonies were picked and mixed with the system. The reaction procedure was started at 94 °C for 5 min, followed by 35 cycles at 94 °C for 30 s, 60 °C for 30 s, 72 °C for 30 s, and ended at 72 °C for 10 min. After amplification, the PCR products were electrophoresed on 1% (*w*/*v*) agarose gel. QIAzol Lysis Reagent (Qiagen, Hilden, Germany) was used to extract total RNA from the spleen samples. After quality control, the mRNA in the total RNA of each sample was separated and fragmented. Double-stranded cDNA was then synthesized and purified. End repair and adaptor ligation were performed on the cDNA. By PCR amplification, the paired-end (PE) library was prepared for sequencing on the Illumina platform (NovaSeq 6000, Illumina, San Diego, CA, USA). The raw sequence data had been deposited in the Genome Sequence Archive (GSA, https://ngdc.cncb.ac.cn/gsa, accessed on 7 October 2023) of China National Center for Bioinformation (CNCB) under the accession number CRA012909.

### 2.8. Identification and Analysis of DEGs

The raw data were filtered to acquire clean reads, which were evaluated by the ratios of Q20, Q30, and GC bases. For subsequent transcript assembly and expression calculation, clean reads were aligned to the mouse reference genome (Accession number: GRCm38). Gene expression levels were calculated based on the number of clean reads located within the genomic region. In addition, the gene expression differences between groups (P18h vs. P0h) were analyzed by using DESeq2 (V1.30.1.). The screening criteria for significantly differentially expressed genes (DEGs) were a false discovery rate (FDR) < 0.05 and |log_2_Fold change (FC)| ≥ 1. GO and KEGG databases were used to analyze the function and related pathways of the DEGs. GO terms and KEGG pathways with corrected *p*-value < 0.05 were considered to be significantly enriched. A heat map was drawn by OmicStudio tools (https://www.omicstudio.cn/tool, accessed on 31 October 2023). STRING (https://cn.string-db.org/, accessed on 22 November 2023) was used to predict the protein–protein interaction (PPI) network, which was visualized through Cytoscape (V3.9.1).

### 2.9. Verification of DEGs by qRT-PCR

Total RNA was extracted from the spleen of mice using the Total RNA Extractor (Trizol) extraction kit (Tiangen Biochemical Technology Co., Ltd., Beijing, China) and then reverse-transcribed into cDNA. To verify the data of RNA-seq, quantitative real-time PCR (qRT-PCR) was performed on the 13 randomly selected DEGs (Table 2). β-actin was taken as an internal reference gene to normalize the transcription level of the target genes. Primer-BLAST (https://www.ncbi.nlm.nih.gov/tools/primer-blast, accessed on 17 November 2023) was used to design specific primers based on their corresponding reference sequences. The design criteria were as follows: (a) The size of the PCR product should be between 60–300 bp; (b) The melting temperature should be 60 ± 2 °C; (c) The primers must span exon–exon junctions. The relative expression levels of the genes were calculated using the formula 2^−ΔΔct^, which represents the ratio of the target gene to the internal reference gene.

## 3. Results

### 3.1. Morphological Characterization and Serotyping of Mh-HN0721

After 24 h of culture on TSA medium, regular, round, and smooth colonies were observed (Appendix A). The microscopic observation showed Mh-HN0721 was Gram-negative and had a reddish, short, rod-like shape (Figure 2a). A single colony was picked from TSA medium and amplified using *M. haemolytica* serotype-specific primers. The results showed that the target band obtained was approximately 160 bp, indicating that the serotype of Mh-HN0721 was A2 (Figure 2b).

### 3.2. Growth Trends of Mh-HN0721

To evaluate the bacterial growth rate and to determine the optimal time point for subsequent experiments, we monitored the OD_600_ values and bacterial quantity from a single colony inoculated in Tryptic Soy Broth (TSB) medium. The results showed OD_600_ value increased from 4 h to 7 h and flattened out after 7 h. The bacterial quantity of Mh-HN0721 probably grew exponentially from 4 h to 5.5 h, reaching a peak value of 3.20 × 10^9^ CFU/mL (Table 3). However, it began to decrease after 5.5 h, indicating that Mh-HN0721 had entered the decline phase.

### 3.3. Pathogenicity and Bacterial Load in Mice Spleen

In the challenge group, mice were depressed, shivering, and hugging at 9 h.p.i, except for two mice that behaved normally and were more active. The body condition of the mice worsened at 18 h.p.i. They exhibited signs such as anorexia, lethargy, and tachypnea. However, as the infection progressed (27 h.p.i), all mice exhibited a restored appetite and alleviated signs. To explore the pathogenicity of Mh-HN0721 on the spleen, the mice were euthanized and their spleens were collected at 9, 18, and 27 h.p.i, respectively. The spleens of the challenge-group mice were all darker red in color and slightly enlarged compared to those of the control group (0 h.p.i) mice. It was noteworthy that the highest bacterial load in the spleen was 9.21 × 10^7^ CFU/g at 18 h.p.i (Appendix A).

### 3.4. Histopathological Observations in Mice Organs

H&E staining showed that there was no significant histopathological change in the heart at 18 h.p.i with Mh-HN0721. At 18 h.p.i., the hepatic sinusoids were dilated. An enlarged white pulp region was observed, along with more lymphocyte apoptosis and necrosis in the spleens at 18 h.p.i. A multifocal hemorrhage in the lungs with increased inflammatory cells in the area of the hemorrhage was observed at 18 h.p.i. There was no obvious histopathological change in the kidney at 18 h.p.i (Figure 3).

### 3.5. Histopathological Changes in Mice Spleen

H&E staining results revealed histopathological changes in the spleens of mice infected with Mh-HN0721 (Figure 4). At 9 h.p.i, the spleens showed a small number of apoptotic lymphocytes and necrocytosis. Comparatively, the most serious histopathological changes was found in the spleens at 18 h.p.i. An enlarged white pulp region was observed, and more lymphocytes underwent karyorrhexis and pyknosis. Moreover, severe apoptosis and necrocytosis appeared in the lymphocytes. In the red pulp, neutrophils and macrophage infiltration occurred, with purplish-red fluid under the splenic tegument. The histopathological changes of spleen at 27 h.p.i were similar to those at 18 h.p.i, but were less severe.

### 3.6. M. haemolytica Identification and RNA-Seq Analysis

To explore the interaction between *M. haemolytica* and the host, RNA-seq was performed on the three spleens at 18 h.p.i. Meanwhile, *M. haemolytica* species-specific PCR was also conducted (Figure 5). A single band appeared around 206 bp, which confirmed that the transcriptomic changes in the spleens were induced by Mh-HN0721. After quality control, a total of 45.92 Gb of clean data were generated from the Illumina platform. The percentage of Q30 bases in each sample was above 94.02% (Appendix A). Differential expression analysis identified 2390 DEGs in P18h vs. P0h (Appendix A), among which 722 were up-regulated and 1668 DEGs were down-regulated (Figure 6a). Statistical differences and expression changes of each DEG are shown in a volcano plot (Figure 6b).

### 3.7. GO and KEGG Enrichment Analysis of DEGs

To screen the key DEGs that related to Mh-HN0721 infection, biological functions of DEGs were annotated and integrated using the GO and KEGG databases. According to GO enrichment analysis, biological process (BP) had the highest number of secondary annotated classifications compared with cellular component (CC) and molecular function (MF). Biological regulation, cellular process, cell part, and binding contained relatively more DEGs than other GO terms in the three ontologies (Figure 7a). For KEGG enrichment analysis, significantly enriched pathways were classified into five categories, including metabolism, genetic information processing, environmental information processing, cellular processes, and organismal systems. The top five pathways containing the most DEGs were signal transduction, immune system, signaling molecules and interaction, endocrine system, and cell growth and death (Figure 7b). Moreover, based on the significance level, the top 20 pathways were shown in Figure 7c. Most of them were signal transduction and immune-related pathways. Based on the expression level, fold change, and significance magnitude, we selected 28 DEGs from these pathways to draw heat maps and construct a PPI network (Appendix A). It showed that the DEGs mainly exhibited an up-regulation trend (Figure 8a). Complex interactions were found among the proteins encoded by the 28 DEGs (Figure 8b), indicating their potential synergistic effects during *M. haemolytica* infection.

### 3.8. Validation of DEGs by qRT-PCR

Thirteen DEGs were randomly selected for qRT-PCR validation of the RNA-seq data. The results showed that *C3*, *Ccl5*, *Cd14*, *Cd34*, *Cd38*, *Cd40*, *Cxcl9*, *Cxcl10*, *Lacm1*, *Itgam*, *Socs3*, *Tnfaip3*, and *Traf1* were up-regulated, all of which were consistent with the RNA-seq results (Figure 9).

## 4. Discussion

The spleen is the largest secondary lymphoid organ and plays a crucial role in the pathophysiology of inflammatory diseases [33]. The spleen is considered a significant source of pro-inflammatory cytokines during the acute phase of the disease and can directly exchange lymphocytes with the bloodstream [34]. Therefore, this paper focuses on the mechanisms of signal transduction and immunoregulation in the spleen following *M. haemolytica* infection in mice. To investigate the immune regulatory mechanisms affected by *M. haemolytica*, we conducted RNA-seq data screening. The enrichment analysis of the sequencing results, which included 2390 DEGs, showed that signal transduction and immune-related pathways were among the top 20 pathways, encompassing 12 of them. These pathways play important roles in the tissue damage caused by *M. haemolytica* infection. According to the KEGG pathway enrichment results, the following pathways were found to be significantly enriched: the NF-κB signaling pathway, cell adhesion molecules, hematopoietic cell lineage, cytokine–cytokine receptor interaction, TNF signaling pathway, IL-17 signaling pathway, chemokine signaling pathway, and several others. Therefore, it may be that the excessive release of cytokines caused by these pathways leads to a severe systemic response.

Research has shown that the NF-κB signaling pathway can impact various cellular processes, including inflammation, immune response, cell proliferation, differentiation, and survival [35]. The results of this study demonstrated that the NF-κB signaling pathway induces an increase in the expression of genes such as the TNF receptor-associated factor *Traf1*, the negative regulatory molecule *Tnfaip3*, and *Socs3*. Traf1 is a unique Traf protein that is predominantly expressed in activated lymphocytes. Traf does not possess any known catalytic structural domains. Therefore, the overexpression of *Traf1* is likely to hinder Tnfr2-mediated apoptosis by altering the composition of the Tnfr2 signaling complex [36]. *Tnfaip3* (also known as *A20*) expresses a cytoplasmic protein that plays a crucial role in the negative regulation of inflammatory and immune responses. Tnfaip3 is a potent inhibitor of the NF-κB signaling pathway and effectively suppresses its activation [37]. The up-regulation of *Tnfaip3* in this paper may contribute to restoring the balance of the NF-κB signaling pathway and preventing excessive signal transduction caused by activation of the NF-κB pathway. Socs3 negative feedback regulates various cytokine signaling pathways and is also a crucial physiological regulator of innate and adaptive immunity [38]. Therefore, Socs3 has a negative feedback regulation on signaling pathways to prevent the overexpression of cytokine signaling pathways, such as the NF-κB signaling pathway, and maintain the stability of the immune system [39]. The expression of these genes may inhibit the NF-κB signaling pathway, reduce the release of pro-inflammatory factors, maintain the balance between pro-inflammatory cytokines and anti-inflammatory cytokines, and ultimately alleviate inflammation in mice. According to the results of RNA-seq, the expression of *Tnfaip3*, *Traf1*, and *Socs3* genes, which are regulated by the NF-κB signaling pathway, was up-regulated and quantitatively verified. The results of qRT-PCR were consistent with those of RNA-seq, confirming the reliability of the sequencing results. Therefore, at 18 h.p.i in the challenge group, it was possible that the NF-κB signaling pathway expressed negative regulatory molecules as a result of negative feedback regulation. This inhibited the expression of the signaling pathway, reduced tissue inflammation, and ultimately enabled the mice to survive.

The cell adhesion molecule pathway may be activated by the NF-κB signaling pathway, leading to an up-regulation of gene expression for chemokines *Ccl5* and *Cxcl10*, as well as the cell adhesion molecule *Icam1*. Ccl5 has been shown to induce lymphocyte aggregation at diseased sites [40]. Other studies have also shown that Ccl5 directly influences the recruitment and activation of T cells and macrophages [41]. Cxcl10 is produced by inflammatory monocytes and fibroblasts in the red pulp of the spleen. This signal from Cxcl10 promotes the differentiation of stem cells [42]. Cxcl10 can promote the proliferation and migration of spleen lymphocytes in vitro [43]. The expression of *Ccl5* and *Cxcl10* significantly increased when bovine turbinate cells were infected with *M. haemolytica* [44]. Icam1 can promote leukocyte migration and is responsible for transporting leukocytes to inflammatory sites [45]. It also induces the infiltration of macrophages and facilitates the adhesion of monocytes and lymphocytes [46]. When bovine bronchial epithelial cells were co-infected with *M. haemolytica* and BHV-1, the up-regulation of *Icam1*, which is associated with leukocyte migration, was observed [47]. At the same time, it was found that the up-regulation of *Itgam* in the cell adhesion molecule pathway also increased the expression of the *Icam1*. This suggests that the pathway of cell adhesion molecules may be associated with cell adhesion and the recruitment of leukocytes. According to the results of RNA-seq, the expression of the *Ccl5*, *Cxcl10*, and *Icam1* genes was up-regulated and quantitatively verified. The results of qRT-PCR were consistent with the results of RNA-seq, confirming the reliability of the sequencing results. In this paper, we found that the NF-κB signaling pathway leads to the up-regulation of the expression of specific cell chemokines and cell adhesion factors. Additionally, it may activate the expression of cell adhesion factors in the cell adhesion molecule pathway. The two pathways collectively promoted the aggregation, adhesion, and differentiation of leukocytes, while also resisting and defending against bacterial inflammation caused by *M. haemolytica*.

The hematopoietic cell lineage pathway may also play a role in the immune system’s resistance to pathogen invasion. The immune cells in the circulatory system continue to migrate into and out of the spleen, even when it is at rest. Additionally, diseases can stimulate the spleen to recruit more cells from other organs [48]. Hematopoietic stem/progenitor cells (HSPCs) may promote cell chemotaxis and adhesion through the involvement of Cd34, thereby inducing the migration and aggregation of HSPCs and facilitating their movement from the bone marrow to the peripheral blood [49]. Therefore, under the influence of inflammation, HSPCs circulating in the blood may be induced to migrate to the spleen for colonization and differentiation. Cd38 is expressed in T cells, dendritic cells, neutrophils, macrophages, and lymphocytes. It promotes cell migration and signaling cascades and is responsible for activating, proliferating, and differentiating multiple immune cells [50]. Cd14 is involved in the clearance of apoptotic cells by macrophages, thereby preventing the release of intracellular substances during cell death and avoiding inflammation and tissue damage. Therefore, the expression of *Cd14* is up-regulated in macrophages infiltrating inflamed areas [51]. Phagocytosis of *M. haemolytica* by phagocytes and CD14 bronchoalveolar lavage cells has been studied [52]. The integrin Itgam plays an important role in leukocyte biology and regulates a variety of leukocyte functions, including cell adhesion, migration, phagocytosis, pro-inflammatory signal transduction, and apoptosis [53]. There are undifferentiated monocyte reservoirs in the red pulp of the spleen, which can be activated to facilitate the rapid recruitment of monocytes in response to tissue damage caused by disease [54]. The sequencing results and quantitative verification in this paper found that the expression of *Cd14*, *Cd34*, *Cd38*, and *Itgam* genes was up-regulated. This may be related to the activation, migration, and differentiation of stem cells into immune cells in the hematopoietic cell lineage pathway.

In summary, this study successfully established a mouse model of *M. haemolytica* infection through intraperitoneal challenge on mice. In our study, we observed that the signs in mice gradually increased after infection and then gradually subsided. RNA-seq analysis of the challenge group at 18 h.p.i revealed a gradual restoration of the mice’s immune system, effectively inhibiting the further development of inflammation. However, our experiment still has limitations. Specifically, we did not continue to detect the expression of spleens when the signs in mice gradually diminished. Therefore, we can only speculate about the process of recovery by observing the ultimate survival of the mice. In addition, the challenge dose in this study may lead to different results at varying doses, which could be another limitation. The consistency of the results can be analyzed by varying the challenge dose across different gradients. Finally, one potential limitation of our experiment is that the adaptation time for mice was only 1 day, which may have still caused stress and increased susceptibility to infection.

## 5. Conclusions

In this study, we successfully constructed a mouse model of *M. haemolytica* infection and observed significant pathological changes in the spleen and other organs. Using RNA-seq, we identified key pathways in the spleen and confirmed their significance through quantitative verification of DEGs in these pathways. This study revealed the mechanism of immune regulation induced by *M. haemolytica* in the spleen of mice. It sheds light on the research about host and *M. haemolytica* interactions and provides a theoretical basis for the prevention, control, and treatment of mannheimiosis.

## Figures and Tables

**Figure 1 animals-14-00317-f001:**
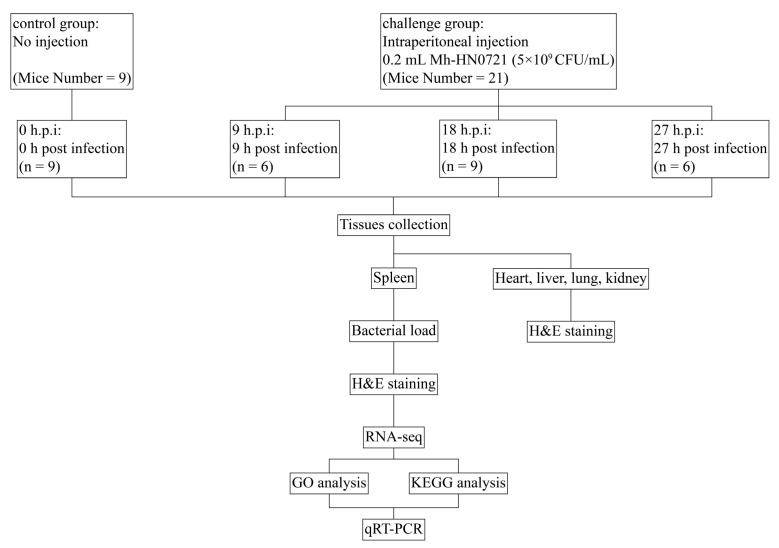
Experimental design of Mh-HN0721 infection in mice. The control group was 0 h.p.i, and the challenge group was divided into 9 h.p.i, 18 h.p.i, and 27 h.p.i. Spleens were collected for bacterial load determination, H&E staining, and RNA-seq.

**Figure 2 animals-14-00317-f002:**
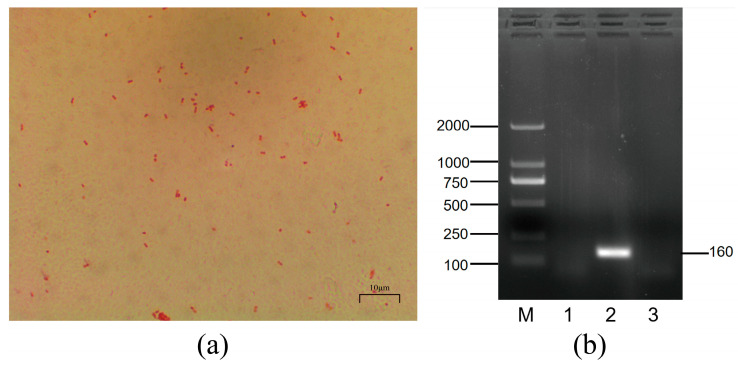
Morphology and serotyping of Mh-HN0721. (**a**) Gram-staining of Mh-HN0721 under the microscope, 100×. (**b**) Serotyping of Mh-HN0721 based on serotype-specific genes. Lane M: DL2000 DNA Marker; Lane 1: A1; Lane 2: A2; Lane 3: A6.

**Figure 3 animals-14-00317-f003:**
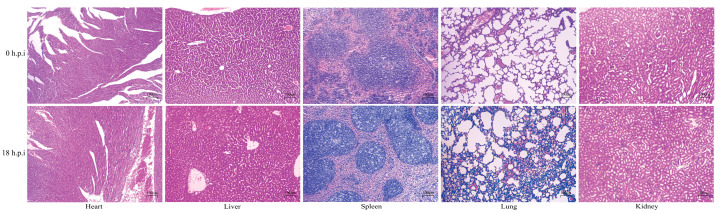
H&E staining of mice organs. Microscopic observation of heart, liver, spleen, lung and kidney in the challenge group (18 h.p.i) and the control group (0 h.p.i) (10×).

**Figure 4 animals-14-00317-f004:**
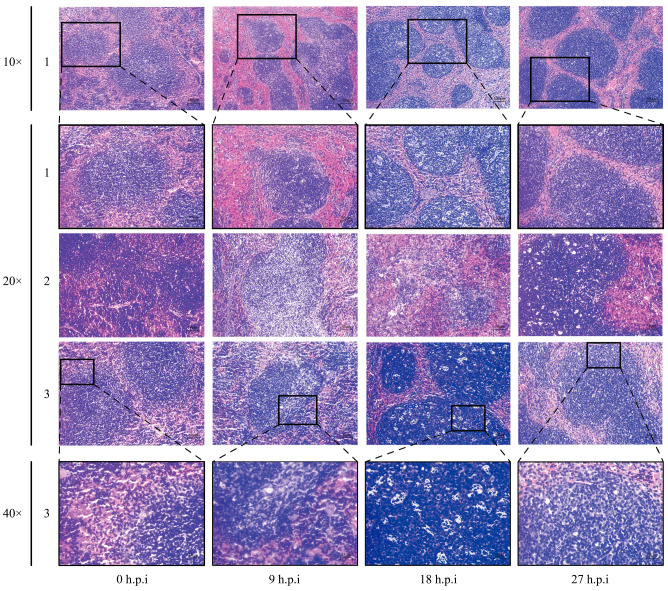
H&E staining of mice spleens in the control group and the challenge group. The numbers of 1 to 3 represent three biological replicates. Stained splenic sections of four time points (0, 9, 18, and 27 h.p.i) are all shown at the magnification of 20×. Samples numbered 1 and 3 are displayed with additional magnifications of 10× and 40×, respectively.

**Figure 5 animals-14-00317-f005:**
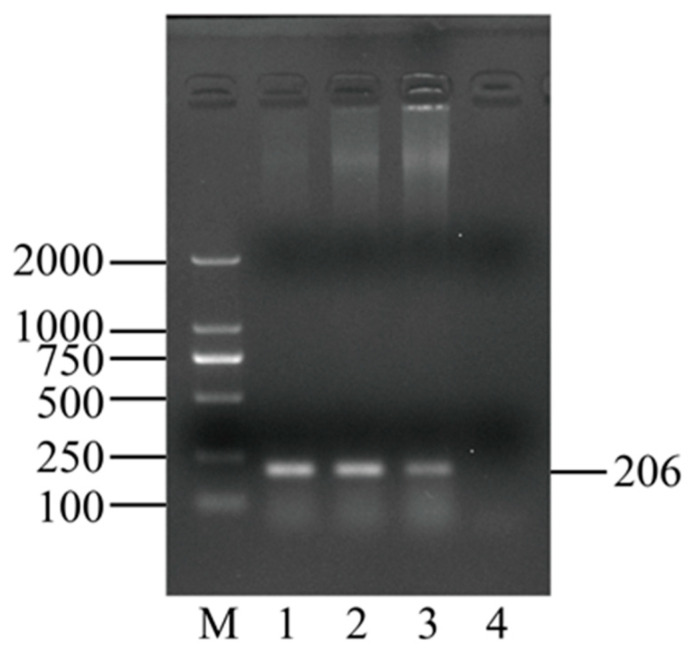
Identification of *M. haemolytica* in the spleens of 3 mice in the challenge group (18 h.p.i). Lane M: DL2000 DNA Marker; Lane 1: DNA of spleen 1 at 18 h.p.i; Lane 2: DNA of spleen 2 at 18 h.p.i; Lane 3: DNA of spleen 3 at 18 h.p.i; Lane 4: blank control.

**Figure 6 animals-14-00317-f006:**
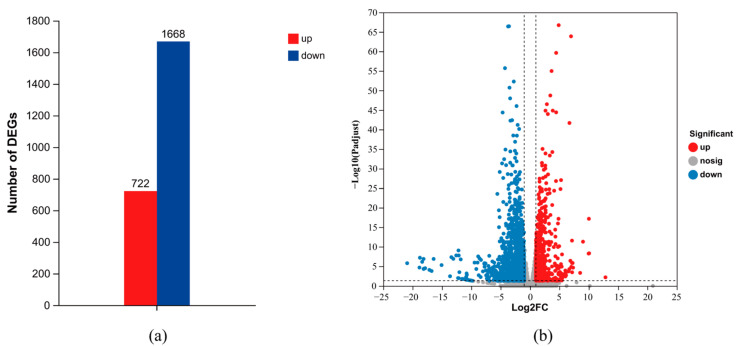
DEGs in P18h vs. P0h (FDR < 0.05 and |log_2_FC| ≥ 1). (**a**) Bar chart of DEGs in P18h vs. P0h. Red represents up-regulated DEGs and blue represents down-regulated DEGs. (**b**) Volcano plot of DEGs in P18h vs. P0h. Red, blue, and gray dots represent up-regulated, down-regulated, and non-significant DEGs, respectively.

**Figure 7 animals-14-00317-f007:**
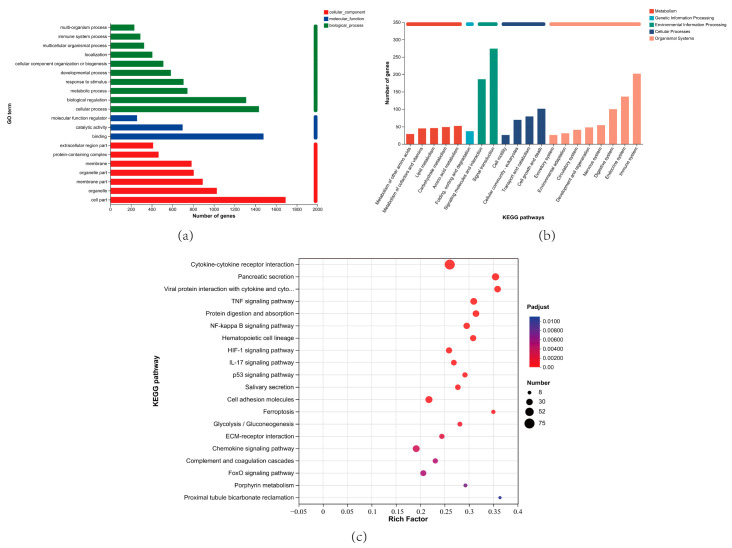
GO and KEGG enrichment analyses of DEGs in P18h vs. P0h. (**a**) GO enrichment bar plot of DEGs in P18h vs. P0h. (**b**) KEGG enrichment bar plot of DEGs in P18h vs. P0h. (**c**) KEGG enrichment scatter plot of DEGs in P18h vs. P0h. Based on DEGs number (**a**,**b**) or significant level (**c**), the top 20 GO terms and KEGG pathways are shown.

**Figure 8 animals-14-00317-f008:**
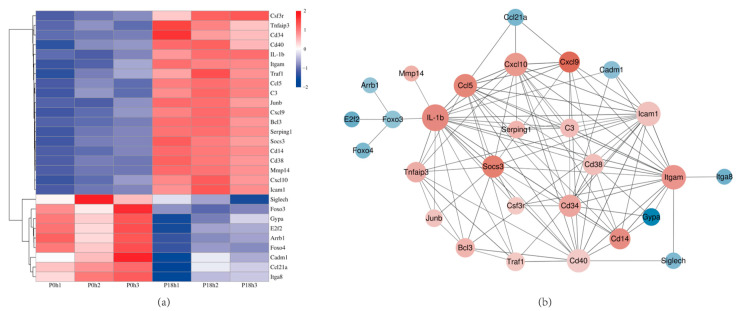
Heat map and PPI network of screened DEGs. (**a**) Heat map of screened DEGs. Color bars from blue to red indicate fold changes of DEGs from decreasing to increasing. (**b**) PPI network of proteins encoded by the screened DEGs. Nodes and edges represent proteins and protein–protein associations. The size of the nodes represents the association degree with other proteins. Up-regulated and down-regulated DEGs are represent by red and blue, respectively. The darker the color, the greater the level of fold changes in DEGs.

**Figure 9 animals-14-00317-f009:**
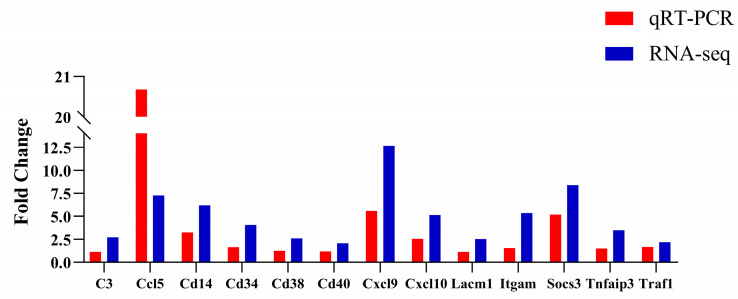
qRT-PCR validation of DEGs. Red and blue represent qRT-PCR and RNA-seq results, respectively.

**Table 1 animals-14-00317-t001:** *M. haemolytica* serotype-specific primers.

Serotype	Primer Sequence (5′–3′)	Product Length (bp)
A1	F: CAT TTC CTT AGG TTC AGC R: CAA GTC ATC GTA ATG CCT	306
A2	F: GGC ATA TCC TAA AGC CGT R: AGA ATC CAC TAT TGG GCA CC	160
A6	F: TGA GAA TTT CGA CAG CAC T R: ACC TTG GCA TAT CGT ACC	78

**Table 2 animals-14-00317-t002:** qRT-PCR genes and corresponding primers.

Name of Genes	Primer Sequence (5′–3′)	Product Length (bp)
*β-actin*	F: GTA CCA CCA TGT ACC CAG GC R: AAC GCA GCT CAG TAA CAG TCC	247
*Cd14*	F: CAC AAT TCA CTG CGG GAT GC R: AGC GAG TTT AGC TGA CTG GG	65
*Itgam*	F: CAG GGC AGG AGT CGT ATG TG R: GTC CAT CAG CTT CGG TGT TG	289
*Cxcl10*	F: TCT CTC CAT CAC TCC CCT TTA R: GCT TCG GCA GTT ACT TTT GTC	151
*Traf1*	F: AGC AGA GGG TGG TGG AAT TAC R: ATC ACG ATG AAG AGG GAC AGG	137
*Tnfaip3*	F: TGG TGT CGT GAA GTC AGG AAG R: CAT TCG TCA TTC CAG TTC CG	251
*Icam1*	F: CAC CGT GTA TTC GTT TCC G R: TGA TCT CCT TGG GGT CCT T	197
*Cd34*	F: CCT TCAG GCT CTG GAA CTC R: TAT AGA TGG CAG GCT GGA CTT	126
*Cd38*	F: GTG GTC CAA GTG ATG CTC R: GTC TAC ACG ATG GGT GCT	286
*C3*	F: CAG CAA CGC AAG TTC ATC R: TTC GCA CTG TTT CTG GTA C	204
*Cd40*	F: CCA ATC AAG GGC TTC GGG TTA R: CTG AGC ACA TGC CTC GCA AT	110
*Cxcl9*	F: CGA GGC ACG ATC CAC TAC AA R: AGG CAG GTT TGA TCT CCG TT	113
*Ccl5*	F: AGG AAC CGC CAA GTG TGT G R: CCG AGT GGG AGT AGG GGA TTA	181
*Socs3*	F: AGA GCG GAT TCT ACT GGA GCG R: CTG GAT GCG TAG GTT CTT GGT C	161

**Table 3 animals-14-00317-t003:** Trends in the growth of Mh-HN0721 over a 12 h period and the concentration of the bacterial solution.

Time (h)	OD_600_ Average Value	Dilution Factor	Number of Colonies	Estimated Concentration of Stock Solution (CFU/mL)
4	0.829	1 × 10^−6^	110	1.12 × 10^9^
			107	
			120	
5.5	1.167	1 × 10^−7^	28	3.20 × 10^9^
			37	
			32	
7	1.235	1 × 10^−6^	35	3.90 × 10^8^
			37	
			45	
8.5	1.221	1 × 10^−5^	135	1.24 × 10^8^
			111	
			125	
10	1.211	1 × 10^−5^	24	2.60 × 10^7^
			22	
			32	

## Data Availability

The datasets presented in this study can be found in online repositories. The raw sequence data have been deposited in the GSA of the CNCB, accession number: CRA012909 (https://ngdc.cncb.ac.cn/gsa, accessed on 7 October 2023).

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
