# Peer review of "Immune Cells in the Spleen of Mice Mediate the Inflammatory Response Induced by Mannheimia haemolytica A2 Serotype"

_animals, 2024, doi:10.3390/ani14020317_

Round 1

Reviewer 1 Report

Comments and Suggestions for Authors

The manuscript entitled “Immune Cells in the Spleen of Mice Mediate the Inflammatory Response Induced by Mannheimia haemolytica A2 Serotype” is focused on the evaluation of the mechanisms of the immunological immune response triggered upon the challenge with M. haemolytica. The obtained results are quite interesting, however, the manuscript required revision. First, it has to be revised and edited by a native/fluent English speaker since there are a lot of grammatical inaccuracies, wrong phrasing, and missing sentences (line 40).

Line 50: the sentence requires revision; M. haemolytica is not a disease. Second, some inaccuracies in data presentation must be corrected. The Authors use phrases like “injection of bacteria” or “treatment with M. haemolytica” while meanwhile it should be used “challenge with bacteria or M. haemolytica”. Moreover, the description of the experimental groups should be “challenge group” and control group” instead of “treatment group” and “blank group”. Table 1 “…. serotyping primers”; it has to be changed into “serotype-specific primers”. What was the age and gender of the mice used in the experiment? Please discuss, why the control group was euthanized at one time and not according to the timetable of the challenge group. The Authors evaluate the bacterial load in the spleen and present data per gram of tissue. However, in the Materials and Methods section, there is no information regarding the weighting of the organs. It has to be included. Line 145: “rpm” should be changed into “g”. Lines 245-248: please, rewrite the sentence, especially change the words “mushroomed” and dwindled”. Please, provide the information on the rationale of using 5 x 109 CFU of M. haemolytica. Moreover, this information should be discussed in the Discussion section. Maybe different infection doses could cause different results.

After providing proper information I recommend the manuscript for publication.

Comments on the Quality of English Language

The manuscript should be edited by a native English speaker

Author Response

Reviewer 1

Comments and Suggestions for Authors

The manuscript entitled “Immune Cells in the Spleen of Mice Mediate the Inflammatory Response Induced by Mannheimia haemolytica A2 Serotype” is focused on the evaluation of the mechanisms of the immunological immune response triggered upon the challenge with M. haemolytica. The obtained results are quite interesting, however, the manuscript required revision. First, it has to be revised and edited by a native/fluent English speaker since there are a lot of grammatical inaccuracies, wrong phrasing, and missing sentences (line 40).

Author response: Thank you for reviewing our manuscript and for the constructive comments, which greatly helped us to improve the manuscript. The manuscript has been reviewed by an experienced editor whose first language is English. We have made revisions to the manuscript in accordance with your comments and highlighted all changes in yellow in the revised manuscript.

Line 40 was changed into “Mannheimia haemolytica (M. haemolytica), previously known as Pasteurella haemolytica, is a facultative anaerobic, gram-negative bacterium that belongs to the genus Mannheimia” on lines 40-42 in revised manuscript.

Line 50: the sentence requires revision; M. haemolytica is not a disease.

Author response: Thank you very much for your correction. Line 50 was changed into “M. haemolytica is an acute infectious pathogen that can cause sepsis and tissue hemorrhaging” on lines 54 in revised manuscript.

Second, some inaccuracies in data presentation must be corrected. The Authors use phrases like "injection of bacteria" or “treatment with M. haemolytica” while meanwhile it should be used “challenge with bacteria or M. haemolytica”.

Author response: Thank you very much for correcting the error. All occurrences of "injection" and "injected" in the manuscript have been changed to "challenge" and "challenged" in lines 28, 153, 155, and 455 in revised manuscript.

Moreover, the description of the experimental groups should be "challenge group" and control group instead of "treatment group" and "blank group".

Author response: Thank you very much for your suggestions. All occurrences of "treatment group" and "blank group" in lines 130, 132, 135, 136, 138, 139, 152, 154, 159, 173, 182, 184, 188, 190, 192, 263, 269, 270, 281, 293, 307, 400, and 457 have been changed to "challenge group" and "control group" in revised manuscript.

Table 1 “…. serotyping primers”; it has to be changed into "serotype-specific primers".

Author response: Thank you very much for your suggestions. Line 122 has been changed from "serotyping primers" to "serotype-specific primers".

What was the age and gender of the mice used in the experiment?

Author response: This experiment using two-week-old male mice. The information has been added to line 124 in the revised manuscript.

Please discuss, why the control group was euthanized at one time and not according to the timetable of the challenge group.

Author response: Thank you for bringing up this question. In response to your concerns, we would like to address the following points:

Our experiments adhered to the ARRIVE 2.0 guidelines[1], and we used the minimum number of mice required, which is in line with animal welfare and ethical guidelines.

The mice were euthanized in chronological order, and the control group was considered as the 0 h.p.i group. We compared the control group with the other groups to study the signaling pathways that change over time. This approach allowed us to analyze the changes in the experimental groups relative to the control group.

We hope this clarifies our experimental design and addresses your concerns.

Reference

[1]ZHANG Junyan, LIU Xiaoyu, LI Yao, et al. Introduction to the International Guide for Animal Research Reporting ARRIVE 2.0, and Its Implementation Plan in the Journal [J]. Laboratory Animal and Comparative Medicine, DOI: 10.12300/j.issn.1674-5817.2023.014.

The Authors evaluate the bacterial load in the spleen and present data per gram of tissue. However, in the Materials and Methods section, there is no information regarding the weighting of the organs. It has to be included.

Author response: Thank you very much for your professional guidance. We added Table A1 in "Supplementary Files". And the reference "(Table A1)" is specifically shown on line 164.

Line 145: “rpm” should be changed into "g".

Author response: Thank you very much for your advice. The “rpm” has been corrected on line 147.

Lines 245-248: please, rewrite the sentence, especially change the words "mushroomed" and dwindled.

Author response: Thank you very much for pointing out the issues in the manuscript. We have deleted these two words. Lines 253-256 were changed into “The bacterial quantity of Mh-HN0721 probably grew exponentially from 4 h to 5.5 h, reaching a peak value of 3.20 × 109 CFU/mL (Table 3). However, it began to decrease after 5.5 h, indicating that Mh-HN0721 had entered the decline phase”.

Please, provide the information on the rationale of using 5 x 109 CFU of M. haemolytica.

Author response: Thank you very much for your question. According to the references [2] and the pre-experiments we conducted, we found that when the dose was too low, the behavioral characteristics of the mice were not noticeable, and we could not determine whether there were pathological changes in the organs. On the other hand, when the dose was too high, the mice died rapidly, and it was difficult to observe the pathological changes completely. Therefore, we chose this particular dose for the challenge.

Reference

  • Lu Biao, Zhang Baohai, Luo Zidan et al.Isolation, identification and complete genome sequence analysis of a strain of Mansoni hemolyticus type 2 isolated from goats [J]. Journal of Yunnan Agricultural University (Natural Science), 2021, 36(04):623-630+699.

Moreover, this information should be discussed in the Discussion section. Maybe different infection doses could cause different results.

Author response: Thank you very much for your suggestions. We have already discussed the limitations of this study in the section of lines 461-464. “In addition, the challenge dose in this study may lead to different results at varying doses, which could be another limitation. The consistency of the results can be analyzed by varying the challenge dose across different gradients”. Finally, thank you again for your careful and professional guidance to our manuscript, which makes me benefit a lot from it!

Reviewer 2 Report

Comments and Suggestions for Authors

The author provided quite details for the M. haemolytica identification and pathogenic data. The results provided some ideas about the pathogen-host interactions and potential prevention and treatment. I have some questions and suggestions as follows:

1. In section 3.3, the author calculated the bacteria load, do you have any raw data like figures of plates or colonies? Otherwise, could you compare the bacteria load via qRT-PCR? You could probably use the serotype or bacterial identification primers to do the qRT-PCR.

2. The Mh-HN0721 was isolated from Hainan black goat and inoculated into a mouse model. Could you provide evidence that all symptoms in the mouse model also happen in infected goats?

3. Figure resolution has to be improved, some figures are quite blurred. Like Figure 1A, Figure 3, Figure 4, Figure 7, and Figure 8.

4. Correct typos in the paper.

Author Response

Reviewer 2

Comments and Suggestions for Authors

The author provided quite details for the M. haemolytica identification and pathogenic data. The results provided some ideas about the pathogen-host interactions and potential prevention and treatment. I have some questions and suggestions as follows:

Author response: Thank you for your kind letter and your careful work regarding our manuscript. We have revised the manuscript in accordance with the reviewers comments.

  1. In section 3.3, the author calculated the bacteria load, do you have any raw data like figures of plates or colonies? Otherwise, could you compare the bacteria load via qRT-PCR? You could probably use the serotype or bacterial identification primers to do the qRT-PCR.

Author response: Thank you very much for your professional question. We have added this information to "Supplementary Files". However, it appears that you have not noticed this information. Therefore, we re-uploaded Table A2 in "Supplementary Files". And we used M. haemolytica identification primers, the sequences of bacterial primer were F: 5’-GCA GGA GGT GAT TAT TAA AGT GG-3’, R: 5’-CAG CAG TTA TTG TCA TAC CTG AAC-3’ is described in line 194 of the original manuscript. Bacteria isolated from the spleen were detected by PCR amplification using M. haemolytica identification primers. The results are shown in Figure 5.

  1. The Mh-HN0721 was isolated from Hainan black goat and inoculated into a mouse model. Could you provide evidence that all symptoms in the mouse model also happen in infected goats?

Author response: Thank you very much for raising this question. The signs observed in the Hainan black goats, including inappetence and listlessness, are similar to those observed in the mice described in lines 263-267. However, differences in signs between the two species, such as coughing and runny nose, may be due to species-specific variations. To confirm this, we plan to conduct M. haemolytica challenge experiments on sheep in our laboratory.

  1. Figure resolution has to be improved, some figures are quite blurred. Like Figure 1A, Figure 3, Figure 4, Figure 7, and Figure 8.

Author response: Dear reviewer, we apologize for uploading figures with low resolution. We have now replaced them with high-resolution figures.

  1. Correct typos in the paper.

Author response: Thank you for your careful review and constructive suggestions regarding our manuscript. We have made revisions to the manuscript in accordance with your comments and highlighted all changes in yellow in the revised manuscript. We appreciate your careful and professional guidance, which has greatly benefited us!

Reviewer 3 Report

Comments and Suggestions for Authors

Dear authors

The study on MH pathophysiologic pathways is interesting and will assist in understanding the lack of treatment response.  The study is relatively well written.  Yet, no recognition of limitations of the sudy at all in the manuscript.  I have suggested to discuss two limitations.  The recognition of limitations does not decrease the value of the study.  It only elevates the value of the manuscript and authors, as they were able to self-reflect.  Self-reflection is important researchers' characteristic.

I have attached PDF with comments.

Comments on the Quality of English Language

My comments are in the attached PDF.

Author Response

Reviewer 3

Comments and Suggestions for Authors

The study on MH pathophysiologic pathways is interesting and will assist in understanding the lack of treatment response. The study is relatively well written. Yet, no recognition of limitations of the study at all in the manuscript. I have suggested to discuss two limitations. The recognition of limitations does not decrease the value of the study. It only elevates the value of the manuscript and authors, as they were able to self-reflect. Self-reflection is important researchers' characteristic.

Author response: Thank you for reviewing our manuscript and for providing constructive feedback that greatly helped us to improve it. We have carefully revised the manuscript and provided a point-by-point response below. We hope that we have accurately addressed all of your comments.

I have attached PDF with comments.

Author response: Thank you again for your patient and professional comments. We have made revisions to the manuscript in accordance with your comments and highlighted all changes in yellow in the revised manuscript.

Line 17: delete " "

Author response: The " " has been deleted on line 17.

Line 40: “also” change "previously"

Author response: “also” was changed into "previously" on line 40.

Line 44: What about primary viral infection?

Author response: Thank you for asking this question. We have conducted extensive research on this topic. The statement has been revised as “M. haemolytica is a part of the natural bacterial community in the upper respiratory tract of ruminants [4,5]. Only transportation, environmental stress, or viral infection can weaken the animal's immune system, rendering it more susceptible to severe respiratory diseases caused by this bacterium, and allowing M. haemolytica to colonize the lungs [6,7].” on lines 46-50 in revised manuscript.

Reference

  1. Santos-Rivera, M.; Woolums, A.; Thoresen, M.; Blair, E.; Jefferson, V.; Meyer, F.; Vance, C.K. Profiling Mannheimia haemolyticainfection in dairy calves using near infrared spectroscopy (NIRS) and multivariate analysis (MVA). Scientific Reports 2021, 11, doi:10.1038/s41598-021-81032-x.
  2. Dao, X.F.; Hung, C.C.; Yang, Y.W.; Wang, J.; Yang, F.L. Development and validation of an insulated isothermal PCR assay for the rapid detection of Mannheimia haemolytica. Journal of Veterinary Diagnostic Investigation 2022, 34, 302-305, doi:10.1177/10406387211068447.
  3. Prysliak, T.; Vulikh, K.; Caswell, J.L.; Perez-Casal, J. Mannheimia haemolyticaincreases Mycoplasma bovis disease in a bovine experimental model of BRD. Veterinary Microbiology 2023, 283, doi:10.1016/j.vetmic.2023.109793.
  4. Griffin, D.; Chengappa, M.M.; Kuszak, J.; McVey, D.S. Bacterial Pathogens of the Bovine Respiratory Disease Complex. Veterinary Clinics of North America-Food Animal Practice 2010, 26, 381-+, doi:10.1016/j.cvfa.2010.04.004.

Line 63: “coloration” change "discoloration"

Author response: “coloration” was changed into "discoloration" on line 63.

Line 65: add "which are"

Author response: we added "which are" on line 65.

Line 72: “breast” change "mammary"

Author response: “breast” was changed into "mammary" on line 72.

Line 74: “by” change "in"

Author response: “by” was changed into "in" on line 74.

Line 75: Several indicate many. please add 1 or 2 more studies of finish the sentence at M. haemolytica.

Author response: Thank you very much for your careful correction! A new reference has been added “[16, 17]” on line 75.

Reference

  1. Kamarulrizal, M.I.; Chung, E.L.T.; Jesse, F.F.A.; Paul, B.T.; Azhar, A.N.; Lila, M.A.M.; Salleh, A.; Abba, Y.; Shamsuddin, M.S. Changes in selected cytokines, acute-phase proteins, gonadal hormones and reproductive organs of non-pregnant does challenged with Mannheimia haemolyticaserotype A2 and its LPS endotoxin. Tropical Animal Health and Production 2022, 54, doi:10.1007/s11250-022-03164-0.

Line 82: “cause” change "facilitate"

Author response: “cause” was changed into "facilitate" on line 82.

Line 85: “strengthen” change "enhance"

Author response: “strengthen” was changed into "enhance" on line 85.

Line 98: Please start the sentence with this portion.

Author response: Thank you very much for your comments. We have deleted the previous sentence and started from this sentence on line 98.

Line 101: “is” change "was"

Author response: “is” was changed into "was" on line 101.

Line 105: extra full stop

Author response: The “.” has been deleted on line 105.

Line 109: “Ethical” change "Ethics"

Author response: “Ethical” was changed into "Ethics" on line 109.

Line 129: 1 day acclimation is insufficient. As it cannot be changed for this study, I would advise this to be discussed as a potential limitation to the study.

Author response: Thank you very much for identifying potential limitations in our experiment. This helped me a lot and improved our experimental design and manuscript at a deeper level. And this issue has been discussed as a potential limitation in the discussion on lines 464-466. “Finally, one potential limitation of our experiment is that the adaptation time for mice was only 1 day, which may have still caused stress and increased susceptibility to infection.”.

Line 160: “points” change "point"

Author response: “points” was changed into "point" on line 160.

Line 165: “conducted serial 10-fold dilutions” change “serially diluted to 10-fold dilutions”

Author response: “conducted serial 10-fold dilutions” was changed into “serially diluted to 10-fold dilutions” on line 165.

Line 174: delete “Three mice in the blank group were euthanized as control.”

Author response: “Three mice in the blank group were euthanized as control.” has been deleted on line 174.

Line 183: “points” change "point"

Author response: “points” was changed into "point" on line 183.

Line 241: “Please add - ‘Supplementary material’ ”

Author response: Dear Reviewer, we apologize for any inconvenience caused by your inability to access the "Supplementary Files". As suggested by Reviewer 2, we realize that we were not clear about this situation. Our original file was uploaded as "Supplementary Files", and a screenshot of the upload is provided below for your reference. To address this issue, we have re-uploaded the "Supplementary Files".

Line 254: “mushroomed” change "probably grew exponentially"

Author response: “mushroomed” was changed into "probably grew exponentially" on line 254.

Line 265: “symptoms” change "signs"

Author response: “symptoms” was changed into "signs" on line 265.

Line 267: “symptoms” change "signs"

Author response: “symptoms” was changed into "signs" on line 267.

Line 274: delete "in the liver"

Author response: “in the liver” has been deleted on line 274.

Line 275: “observed” change “observed, ”

Author response: “observed” was changed into “observed, ” on line 275.

Line 277: “hemorrhage” change "hemorrhage was observed"

Author response: “hemorrhage” was changed into "hemorrhage was observed" on line 277.

Line 336: Probably single figure per page (meaning one page Fig 7 and other page Fig 8).  Currently, text in Figs not readable

Author response: Thank you very much for your typesetting suggestions. Figure 7 and Figure 8 have been put on a separate page, and all the pictures have been replaced with high-resolution pictures on line 335.

Line 456: “symptoms” change "signs"

Author response: “symptoms” was changed into "signs" on line 456.

Line 460: Another limitation that should be mentioned is that you do not have spleen expression at the moment of decrease.

Author response: Thank you very much for your advice. This is a great help to our manuscript! Your suggestion made me realize that these limitations do not reduced the value of the study. The limitation have so far been written into the discussion on lines 459-462. “However, our experiment still has limitations. Specifically, we did not continue to detect the expression of spleens when the signs in mice gradually diminished. Therefore, we can only speculate about the process of recovery by observing the ultimate survival of the mice”. Finally, thank you again for your careful and professional guidance to our manuscript, which makes me benefit a lot from it!